# Extracellular Vesicles as Innovative Tool for Diagnosis, Regeneration and Protection against Neurological Damage

**DOI:** 10.3390/ijms21186859

**Published:** 2020-09-18

**Authors:** Pavle Andjus, Maja Kosanović, Katarina Milićević, Mukesh Gautam, Seppo J. Vainio, Denis Jagečić, Elena N. Kozlova, Augustas Pivoriūnas, Juan-Carlos Chachques, Mirena Sakaj, Giulia Brunello, Dinko Mitrecic, Barbara Zavan

**Affiliations:** 1Center for Laser Microscopy, Faculty of Biology, University of Belgrade, 11000 Belgrade, Serbia; pandjus@bio.bg.ac.rs (P.A.); kmilicevic93@gmail.com (K.M.); 2Institute for the Application of Nuclear Energy, INEP, University of Belgrade, 11080 Belgrade, Serbia; maja@inep.ac.rs; 3Feinberg School of Medicine, Northwestern University, Chicago, IL 60611, USA; mukesh.gautam@northwestern.edu; 4Faculty of Biochemistry and Molecular Medicine, Disease Networks Research Unit, InfoTech Oulu, Kvantum Institute, University of Oulu, Borealis Biobank of Northern Finland, Univ. Oulu Hospital, FI-90014 Oulu, Finland; seppo.vainio@oulu.fi; 5Laboratory for Stem Cells, Croatian Institute for Brain Research, University of Zagreb School of Medicine, HR-10000 Zagreb, Croatia; Denis@gmail.com (D.J.); Mitrecic@gmail.it (D.M.); 6Laboratory of Regenerative Neurobiology, Department of Neuroscience, Uppsala University, 75237 Uppsala, Sweden; elena.kozlova@neuro.uu.se; 7Department of Stem Cell Biology, State Research Institute Centre for Innovative Medicine, LT-01001 Vilnius, Lithuania; augustas.pivoriunas@imcentras.lt; 8Laboratory of Biosurgical Research (Alain Carpentier Foundation), Pompidu Hospital, University Paris Descartes, 75015 Paris, France; j.chachques-ext@aphp.fr; 9Department of Molecular Science and Nanosystems, Ca’ Foscari University of Venice, 30100 Venice, Italy; mirenasakaj@gmail.com; 10Department of Neurosciences, University of Padova, 35128 Padova, Italy; giulia.brunello@unipd.it; 11Department of Morphology, Surgery and Experimental Medicine, University of Ferrara, 44121 Ferrara, Italy

**Keywords:** exosomes, microvesicles, drug delivery, neurological disorders, theragnostic blood–brain barrier, stem cells, neuronal development, miRNA

## Abstract

Extracellular vesicles (EVs) have recently attracted a great deal of interest as they may represent a new biosignaling paradigm. According to the mode of biogenesis, size and composition, two broad categories of EVs have been described, exosomes and microvesicles. EVs have been shown to carry cargoes of signaling proteins, RNA species, DNA and lipids. Once released, their content is selectively taken up by near or distant target cells, influencing their behavior. Exosomes are involved in cell–cell communication in a wide range of embryonic developmental processes and in fetal–maternal communication. In the present review, an outline of the role of EVs in neural development, regeneration and diseases is presented. EVs can act as regulators of normal homeostasis, but they can also promote either neuroinflammation/degeneration or tissue repair in pathological conditions, depending on their content. Since EV molecular cargo constitutes a representation of the origin cell status, EVs can be exploited in the diagnosis of several diseases. Due to their capability to cross the blood–brain barrier (BBB), EVs not only have been suggested for the diagnosis of central nervous system disorders by means of minimally invasive procedures, i.e., “liquid biopsies”, but they are also considered attractive tools for targeted drug delivery across the BBB. From the therapeutic perspective, mesenchymal stem cells (MSCs) represent one of the most promising sources of EVs. In particular, the neuroprotective properties of MSCs derived from the dental pulp are here discussed.

## 1. Introduction

Communication between cells plays an essential role in the survival of every multicellular organism, by providing coordination of all processes that take place in it, at the intercellular, tissue, organ and interorgan level. Until recently, it was believed that only two types of intercellular communication existed: by direct contact of molecules on surfaces of adjacent cells and by secreting soluble molecules by one cell that reached neighboring or distant cells, thus affecting their behavior. However, in the mid-2000s it was discovered that one cell can also send to another a signaling package of molecules, wrapped in biological membrane–extracellular vesicles (EVs) [1]. EVs are small, usually round, structures, from 30 to 1000 nanometers in size, which can transport all kinds of biological molecules to target cells (proteins, non-coding RNA, mRNA, lipids, etc.) and are involved in all physiological and pathophysiological processes studied so far [2]. “EVs” are actually a broad term for a heterogeneous population of vesicles differing in their mode of biogenesis, size, cellular origin, composition and roles [3].

According to mode of biogenesis there are two major types of EVs: exosomes, which are of endosomal origin, and microvesicles (MVs), which bud out from the plasma membrane. Apoptotic bodies are also considered as EVs in a broader sense, but their role in communication is less studied and will not be considered here [2,3]. Exosomes originate in late endosomes, by inward budding of their membrane, either by an endosomal sorting complex required for transport (ESCRT)-dependent or ESCRT-independent mechanism [3,4]. Both processes comprise a selection of cargo molecules and result in the formation of a multivesicular body (MVB) containing multiple intraluminal vesicles (ILVs). The ESCRT-dependent mechanism involves selection and transport of ubiquitinated proteins into invagination of the endosomal membrane, which is subsequently released into the lumen of MVB as ILV, all dependent on the orchestrated action of members of the ESCRT complex [5,6,7]. As for ESCRT-independent mechanisms, tetraspanins, Hsc70 and sphingosine-1-phosphate are shown to be involved in protein cargo selection [8,9,10]. In addition, lipid rafts are presumed to play a role since their components are found in EVs [11]. Ability to induce the membrane curvature and thus form ILV was described for ceramide (precursor of sphingosine-1-phosphate) and phosphatidic acid [12,13]. Upon formation of ILV, MVB is targeted to the plasma membrane by RAB proteins [3,14,15,16]. Different RABs are responsible for guiding MVB with different types of exosomes, as reviewed in [3]. Finally, N-ethylmaleimide-sensitive fusion attachment protein (SNAP) receptors (SNARE) were shown to be involved in MVB and plasma membrane fusion resulting in the release of ILV into extracellular space as exosomes [17]. Biogenesis of MVs, on the other hand, comprises of Ca^2+^ influx dependent activation of enzymes that disassemble the cytoskeleton at the MV formation site and ones that change the lipid composition of the membrane, allowing its reshaping and outward budding until final MV detachment [18]. Specific members of the ESCRT complex may be involved in this process, as well [19,20].

Despite these different biogenesis pathways there is still no definitive markers for distinguishing exosomes from MVs. Exosomes lack common glycolytic enzymes and cytoskeletal elements; suggesting that exosome loading must be a highly regulated process. Based on several evidence it is clear that EVs are a heterogenic family related not only to the differences in isolation, but also to concentration and purification strategies. Much of the recent interest in EVs was triggered by the discovery that exosomes function in the transport of secreted extracellular RNA (exRNA), in Argonautes (Agos) that are important miRNA-processing proteins, in RNA- binding proteins (RBPs) and in an extracellular receptor.

Although some proteins appear to be enriched in some EVs types, such as tetraspanins CD63, CD9 and CD81 in exosomes, there is a significant overlap of their presence in different types and subpopulations of EVs so these proteins cannot be defined as distinction markers [21]. There are three main reasons for the lack of such markers: there is no molecule shared between all exosomes or all MVs, as far as currently known; both exosomes and MVs actually comprise of different subpopulations, whose biogenesis, sorting of cargo molecules and types are still poorly defined; and the most frequently used isolation procedures do not yield ‘pure’ subpopulations, so most findings refer to the mixture of EV types and their subpopulations [21,22,23]. Aside from different modes of biogenesis, exosomes and MVs also differ in their average size. However, although exosomes are generally smaller (30–250 nm) compared to MVs (200–1000 nm), significant overlap exists, so size too cannot be used for their distinction [24]. Given that exosomes and MVs still cannot be distinguished or completely separated, either by markers or size, the International Society for Extracellular Vesicles (ISEV) recommends the term EVs to be used [25,26]. It should be kept in mind that the terms EVs and exosomes are sometimes used as synonyms in the literature, hence the isolation procedure needs to be emphasized.

EVs, generally, comprise of molecules involved in different cellular processes: EVs biogenesis, membrane organization and trafficking, adhesion, membrane fusion, signal transduction, metabolism, etc. [27]. However, the composition of a particular EV depends mainly on its cellular origin and type. EVs of the same type may share components related to the mode of biogenesis. Thus, exosome content includes proteins of the ESCRT complex, RAB proteins, tetraspanins and some lipid-raft associated proteins and lipids, while MVs are rich in plasma membrane components, specifically lipid raft content, cytoskeleton networks associated proteins, phospholipase PLD2 and specific ESCRT members [3]. Nevertheless, the content of EVs is also related to their subpopulations. Thus, Kowal et al. define several large and small subpopulations of EVs and component groups differing or overlapping between those subpopulations [23]. As for components of EVs related to their origin, they comprise of specific cell surface receptors, enzymes or markers specific to the cell type [27]. Lists of components found in particular EVs can be found in public databases: Vesiclepedia, EVpedia or ExoCarta. It is important to note that EV components are not randomly encapsulated molecules from the originating cell but rather selectively packed. The selection of proteins depends on biogenesis machinery, while selective sorting of RNA, specifically miRNA, depends on their sequence, cellular abundance and competition with their targets. Mechanisms of, e.g., miRNA sorting rely on the interaction with RNA-binding proteins, which helps loading RNA via protein sorting mechanisms [3]. Additionally, in consideration of EV composition it should be kept in mind that besides ’true’ EV components (lumen and membrane ones) EVs can also transport components associated with its surface [3,23]. Once released, EVs transport their content towards near or distant target cells, i.e., they can act in an autocrine, paracrine or endocrine manner [28]. Exosome exerts its versatile effects on fundamental biological processes in pleiotropic manners, such as directly activating cell surface receptors via protein and bioactive lipid ligands, fusing their membrane contents into the recipient cell plasma membrane and delivering effectors including transcription, oncogenes, small and large non-coding regulatory RNAs into recipient cells [29,30,31,32,33,34,35,36,37,38,39,40,41]. In order to exert their role, i.e., transfer information, EVs can interact with the target cell in one of the following three ways: by the direct contact of molecules on surfaces of EVs and the plasma membrane of the target cell (receptor–ligand interaction); EVs can be taken up by endocytosis; or EVs can fuse with a plasma membrane and release their content in the cytoplasm of the target cell while their membrane becomes part of the target cell plasma membrane [2,42,43].

The physiological roles of EVs extend to all physiological processes investigated so far. Thus, EVs are implicated as important players in coagulation, reticulocyte maturation, innate and acquired immunity, placentation, embryonic development, tissue repair, bone calcification, liver homeostasis, nervous system function, etc. [2]. Moreover, EVs are involved in many pathological processes, such as cancer, cardiovascular diseases and immune disorders, diabetes, obesity, atherosclerosis, neurological disorders, etc. [44,45].

Due to their unique properties and involvement in pathological processes, EVs hold great potential to be exploited in diagnosis and therapy. Since the cargo of EVs is a complex selection of molecules from the cell of origin, it constitutes a representation of the cells’ status at the time of release, providing a valuable basis for using EVs in diagnostics [46]. EVs are found in all body fluids, so they are easily accessible and thus called “liquid biopsies” [47]. This is especially important for the diagnostic of tissues not easily accessible by biopsy. Additionally, EVs cross the blood–tissue barriers, including blood–brain, and are stable in body fluids [48,49]. In different pathologies, EVs can change in number, size or biomarker (protein, RNA and lipid) content [50,51], so these features may be used for designing new diagnostic tests. The potential of EVs in diagnostics was suggested for cancers (prostate [52], glioblastoma [53], breast [47], lung [54], bladder cancer [55], etc.), infections [56], neurodegenerative [57] and other diseases.

In addition, EVs have excellent capacity to yield new therapeutic approaches: they cross the blood–tissue barriers, they are biocompatible, they have natural targeting, which could eventually be further manipulated by bioengineering, and they can be loaded with drugs [46,58]. One approach, already in the clinical trial pipeline, is to use EVs as vaccines to enhance the immune response to tumors (specifically antigen presentation), by EVs from dendritic cells treated with tumor antigens [59]. The same approach can be applied for the treatment of infectious diseases [60,61]. EVs can also be used to suppress immunity in graft-vs-host disease or in autoimmune diseases, as shown for EVs from mesenchymal cells or parasites [62,63]. Finally, EVs can be used for (synthetic or biological) drug delivery [64,65]. Although there are significant benefits of EV-based therapies, such as a lack of tumor-inducing risk or lowering overall toxicity of loaded drugs, there are some issues that need to be addressed before these approaches can be fully exploited. We need to identify active molecules in different therapeutic EVs, set standardization and quality control, explore the need for adjuvants, the loading efficacy and the possibility to use synthetic vesicles loaded with active and targeting molecules instead of isolated EVs [46]. As all these issues are being addressed, EV-based therapy is expected to considerably advance the treatment of many diseases. Furthermore, since EVs hold great potential for both diagnostic and therapy, they might be used in “theranostics” to both identify targets in disease and to deliver therapy [46].

Embryonic development is critically based on cell–cell and tissue interactions. During this process the progenitor cells with the same or different history exchange signals that direct gradual differentiation of the cells towards specific lineages. The signals secreted by the cells direct the cell behavior in many ways and morphogenesis involves such events as cell division, changes of cell shape, cell migration and controlled cell death via apoptosis, the programmed cell death. Collectively the cellular level process directs the spatial and temporal arrangement of the cells shaping embryonic development [47,48,49,50,51].

Of the developmental steps, the process of gastrulation represents the most critical one. During gastrulation the rather undifferentiated epithelial cells start to invaginate, thus they invade under the ectoderm and alongside of this generate the mesodermal germ cell layer. Gastrulation involves an epithelial to mesenchymal transition that is connected to the formation of the mesodermal embryonic tissue layer [52,53,54,55,56,57,58,59,60,61]. At the same time in the ventral embryonic part the endoderm becomes formed.

The process of gastrulation is crucial since it is a process where the embryonic tissue layers form. Here the prospective anterior position of the emerging body plan serves as the organizer. The organizer secretes signals across the mesodermal cells, and these go on to pattern cell fates along the dorsal and ventral and anterior posterior embryonic axes [63,64,65,66,67,68,69,70,71,72]. Additionally, in association to this positional specification defined mesodermal segments induce the overlying ectoderm to differentiate into neural tissue [72,73,74,75,76,77].

For a long time, the signals that regulate morphogenesis and also associate with the fundaments of the generation of the main embryonic signaling center, the organizer, were enigmatic. However, when the growth factor families were identified and studied in the embryonic phenomena it was found that unexpectedly relatively few secreted growth and differentiation factors were mediating the early mesoderm and neural induction. Here, as shown in Figure 1, the key players were the Wnts, FGFs, Hedgehogs and TGF beta family members and antagonists of these pathways [78,79,80,81,82,83,84,85].

A key concept in developmental cell–cell signaling is that the inductive cells, such as the ones from the organizer, are establishing gradients. Yet, in partly open signaling processes morphogens are speculated to be secreted by the inducing tissue, establishing a gradient in the responding tissue, which governs the process of morphogenesis. This process yields the detailed spatial and temporal organization of the tissue, as in the primary embryonic induction of the central nervous system development. However, the process of induction is critical not only in early developmental steps but also later, during organogenesis [86,87,88,89]. The later inductive processes are named as the secondary inductive tissue interactions. As is the case for the primary induction, the secondary inductive signaling occurs via the depicted core developmental signaling cassette. The key signals that regulate organ morphogenesis are also involved later, in the adult in the maintenance of the stem cell niches for renewing organs such as skin, liver and hematogenic tissue, and in regeneration from tissue injury.

The experimental evidence from knockout and ex vivo explant organ culture and tissue recombination assays are however not fully in line with the notion that the growth factors, such as the Wnts, would serve as the morphogens, to increase distances from the origin of their secretion forming a gradient that goes on to determine cell fates via the developmental genomic program. Given these open issues the secreted nano- and microvesicles and of these the exosomes provide a relevant novel opening and are good candidates for the long-searched morphogens [90,91,92,93].

Since the exosomes are secreted from cells similarly as the growth factors and contain a wealth of signaling molecules as their molecular cargo (including DNA, RNA, metabolites and proteins), they can be considered excellent morphogen candidates. The property of the exosomes to be taken in by a target cell via a receptor-mediated mechanism and to deliver their content to a target is in line with complex signaling factors expected from critical cell fate regulators as well.

At the time being, evidence starts to be available that the EV is present and secreted by embryonic cells and that are associated with tissue interactions where the morphogenetical active cell signaling occurs. The presence of a wealth of enzymes, RNA and protein species fit well to the concept of morphogen by serving as developmental gene circuit programmers and possibly as epigenetic and even transgenerational regulators [94,95].

The mammalian kidney serves as a good model to study the exosome roles in inductive cell–cell signaling. Evidence is available that the ureteric bud cells that normally trigger the nephrogenesis program are indeed secreting the exosomes. The ureteric bud cell-derived exosomes can even enter the nucleus of the target nephrogenic mesenchyme. Embryonic kidney exosomes carry many proteins including the Wnt pathway components. The exosomes secreted from the inducer tissue are present in the interphase of the interactive cells.

When purified exosomes are exposed to an embryonic kidney mesenchyme, which has the competence to form the nephrons, such exosomes induce condensation, but not yet a full-blown activation of the nephrogenesis program. The fact that purified exosomes do not, in the current experimental setting, activate the full developmental program is in line with the earlier conclusions that direct cell–cell contacts are required for successful transmission of the nephrogenesis program activation. Thus, it may be that close cell–cell contacts are required to coordinate inductive signaling to the dynamics of cell divisions, cell shape changes and migration that associate with the morphogenetic field where the exosomes are robustly present. Thus, further sophisticated ways to edit the exosome molecular cargo, single cell transmission and imaging are likely needed to reveal the detailed cellular and molecular mode of action of the exosomes in the inductive or morphogenetic cell and tissue interactions that govern embryogenesis including organogenesis. In summary the exosomes are excellent candidates for local and more long-term developmental signals, but further studies are needed to provide conclusive evidence for this [96,97].

## 2. Type and Role of EVs

### 2.1. EVs in Transplacental Communication

One of the blood–tissue barriers that EVs are known to cross is placenta. Placenta is a specific type of barrier in which maternal blood comes in direct contact with semiallogenic tissue. Placental villi, which anchor the placenta to the uterine wall, consist of blood vessels surrounded by connective tissue, which is covered by two cellular layers, cytotrophoblast and syncytiotrophoblast. Syncytiotrophoblast is immersed in the mothers’ blood present in intervillous, lacunar space. On the mothers’ side, blood vessels pass through a modified endometrial tissue of the uterus, decidua and open to lacunae [96]. This architecture of placenta allows oxygen and nutrients to reach the fetus but it also provides controlled exchange of information between mother and fetus. This exchange is especially important for establishing a balance between the immune tolerance to semiallogenic fetal tissues and maintenance of the ability to exert immunity towards pathogens. One of the most important carriers of information in feto–maternal cross-talk throughout pregnancy are EVs, which are produced by several cell types of both mother and fetal origin and transfer information in both directions [97].

Thus, it is found that syncytiotrophoblast (STB) produces EVs, which carry placental miRNA, are released in the mothers’ blood and can be found in the mothers’ circulation [98]. The level of STB EVs increases during pregnancy [99] and can constitute 12–25% of the total EVs in the mothers’ blood [100]. The level of STB EVs is related to the mothers’ body mass index [100] and some pathologies, as gestational diabetes mellitus [101]. Bidarimath et al. (2017) demonstrated bidirectional shuttling of miRNA carrying EVs between trophectoderm and the mothers’ endothelial cells and their influence on each other’s proliferation [102]. STB EVs were found to increase proinflammatory cytokine production in human umbilical cord vein endothelial cells, HUVEC [101]. However, more detailed analysis has shown that trophoblast produces two types of EVs: smaller exosomes and larger microvesicles (MV), which are both found in maternal circulation [103] and possibly have distinct roles. Exosomes carry molecules (FasL and TRAIL), which provide tolerance towards the fetus by inducing apoptosis in the mothers’ T lymphocytes and activated mononuclear cells [104,105] or downregulate the cytotoxicity of NK, CD8+ and T cells (by downregulating NKG2D with ULBP1–5 and MIC ligands) [106]. Placental exosomes can also suppress T cell proliferation by specific miRNA [107]. However, STB exosomes were also reported to have proinflammatory properties as they can recruit monocytes [108]. In addition, they have antiviral properties as well [109]. On the other hand, placental microvesicles were shown to influence cytokine production by the mother’s B cells and promote a Th2 type of immune response [110]. Similar to STB exosomes, STB MVs were also shown to have a proinflammatory function by activating monocytes [111].

In addition to the exchange of EVs produced locally at the feto–maternal interface, there is also evidence that non-placental EVs can be transferred trough the placenta, in both directions. Thus, EVs from amnion epithelial cells, secreted in amniotic fluid, can pass through and reach the maternal side of the placenta and maternal blood, uterus and kidneys, as evidenced by intra-amniotic injection of labeled human amniotic epithelial EVs into pregnant mice [112]. On the other hand, a finding in the placenta and other fetal tissues, including the heart, of labeled circulating EVs from other pregnant mice injected in the tail vein shows transplacental mother-to-fetus EVs transfer as well. In addition, these EVs, when isolated from diabetic mice, can contribute to the cardiac development deficiency in the fetus of healthy mice [49].

These findings suggest that EVs can cross the placental barrier and eventually be used as diagnostic and therapeutic tools.

### 2.2. Microglial EVs—The Good and the Bad

Microglia are resident mononuclear phagocytes in the central nervous system (CNS), which are responsible for maintaining tissue homeostasis and immune defense. During embryonic development their precursors migrate from the yolk sac to the CNS where they differentiate into microglia [113]. In the resting state of the adult CNS, these cells constantly monitor their environment and interact with other cell types in order to maintain tissue homeostasis and plasticity [114,115]. These cells are equipped with numerous receptors (i.e., purinergic, for the complement system, immunoglobulins, adhesion molecules and inflammatory stimuli), whose activation leads to the release of neurotrophic factors, pro- and anti-inflammatory cytokines [115,116].

EVs represent important means of communication between the periphery and CNS. Microglia are a good example of cells behaving both as targets and sources of EVs. The circulating EVs have a potential to act as neuroinflammatory mediators [117]. Intravenously administered serum-derived exosomes from lipopolysaccharide (LPS) challenged mice to healthy mice induce microgliosis and to a lesser extent astrogliosis, followed by increase in expression of pro-inflammatory cytokines and microRNA [117]. On the other hand, microglia themselves release EVs under resting conditions [118]. However, during inflammation these cells increase the amount and change the content of released EVs [119,120,121,122], as evidenced by exosomal markers CD63 and flotilin-2 in the culture medium lysates from IFN-γ/LPS-treated midbrain slice cultures [119]. A similar exosomal response of BV-2 cells was induced by α-synuclein (expressed as an increased production of MHC class II molecules and membrane TNF-α; [120]). Upon stimulation of BV-2 microglial cells with LPS the release of EVs is changed in terms of size (they secrete larger EVs in a higher number) and content (increased IL-6 and TNF-α and distinct proteomic profile with 49 proteins specific for LPS-induced EVs, with some of them with an importance in neurodegeneration; [122]). In a model of neuroinflammation, in mice subjected to permeant middle cerebral artery occlusion, inhibition of TNF-α signaling pathway results in a robust reduction in the number of vesicles released by microglia both in vitro and in vivo [122], thus underlining that the mechanism of EV release is a highly regulated process that calls to be studied in more detail.

EVs released by microglia during inflammation can have detrimental effects on neurons, such as the decreased number of dendritic spines on hippocampal neurons [123] or even cell death in dopaminergic neurons [119]. These data indicate a direct role of microglia-derived EVs in neurotoxicity and neurodegeneration. Moreover, exosomes released by activated microglia can induce the activation of astrocytes [124] or resting microglia [125], suggesting that microglia-derived EVs can influence different cell types in the CNS and promote inflammation. Microglia-derived EVs can help spread neurotoxic molecules such as Aβ [126] or tau protein [127]. In addition, EVs are not just carriers, but they are also biologically active—they can actively transform insoluble Aβ aggregates to neurotoxic forms important for the propagation of Alzheimer’s disease [126].

Although there is a great deal of evidence that microglia-derived EVs have an important role in promoting inflammation and they have a role in CNS repair too. In acute states, such as the traumatic brain injury, microglial exosomes are enriched in miRNA that promotes the M2 state of other microglial cells, inhibit neuronal injury and stimulate neurite outgrowth [128]. Additionally, microglial cells internalize neuronal-derived exosomes carrying Aβ and degrade them inside lysosomes [129]. These studies demonstrate the apparent importance of microglia with EVs in determining the progression of neuroinflammation and neurodegeneration. On the other hand, it is also important to emphasize that microglial release of EVs is a valuable homeostatic mechanism in the resting state. Some of the known functions of these EVs are metabolic support for neurons [118] and local regulation of neuropeptide activity [130]. Furthermore, EVs shed from microglia have an important role in modulating neuronal activity by interacting with neuronal plasma membrane receptors and enhancing spontaneous and evoking excitatory transmission without producing any acute or delayed neurotoxic effect [131]. Interestingly, astrocyte-derived EVs have the same potential, however with lesser efficiency [131].

In conclusion, EVs are of great significance for microglial communication with other cell types within the CNS as well as with the periphery with the advantage of coordinated regulation of processes over long distances and not just intracellularly. Depending on their content they can either be regulators of normal CNS function and repair or propagators of neuroinflammation and neurodegeneration.

### 2.3. EVs and Neural Stem Cells

Neural stem cells (NSCs) are multipotent cells that are capable of differentiating into the three major CNS cell types: astrocytes, oligodendrocytes and neurons. They became almost an obligatory part of the neuroscience toolbox, so they are used in basic research [132] and in new translational strategies [133,134]. Their biological properties mostly depend on secreting intrinsic and extrinsic factors and molecules by which they control neural development and recovery [135]. These factors and molecules include: growth factors, proteins, miRNA and vesicles, which when secreted are known as extracellular vesicles. Extracellular vesicles not only control stem cell properties, but also by initiating intercellular cross-talks, they modify properties and the functionality of distal cells of the CNS [136].

NSC-derived EVs express strong neuroprotective, neuroregenerative, neuroproliferative and antiaging properties [137,138]. As Sun et al. [139] suggested exosomes play a crucial role in neurorestoration after ischemic injury in vitro. When exosomes were applied to primary cell culture of astrocytes and neurons prior and after oxygen-glucose deprivation (OGD), they protected and induced the survival of astrocytes. This result was also confirmed in vivo where the stroke volume was significantly decreased after intravenous (IV) injection of exosomes following middle cerebral artery occlusion (MCAO) [139]. Similar results were also observed in the porcine model after MCAO treatment. After IV injection of EVs, significant improvements were seen in the form of a reduced loss of brain affected area, in preservation of white matter integrity and decreased brain edema [140]. Interestingly, the method of obtaining stem cells influences the types of exosomes they secrete. Thus Ma et al. [141] showed that induced exosomes (iEXOs) secreted from induced neural progenitor cells (iNPCs) stimulated the proliferation of neural progenitor cells (NPCs), as opposed to exosomes secreted by NPCs, which only possess a differentiating ability. This effect was mediated by miR-21, which is highly expressed in NPCs and in later stages of brain development [141]. NSCs’ exosomes and their mi-RNA also have an important role in the process of aging. The study by Zhang et al. [138] confirmed that the aging process largely depends on the number of NSCs in the hypothalamic area of the brain, which drastically decreases over time. Selective removal of these cells using an injection of lentivirus expressing herpes simplex virus leads to cognitive and memory decline, coordination disbalance and acceleration of the aging process shortening lifespan. On the contrary, postnatal transplantation of NSCs in the region of the hypothalamus slows down aging and increases longevity [138]. During neonatal development NSC-derived exosomes from the subventricular zone (SVZ) interact with microglia. Let-7 miRNA stimulates the release of proinflammatory IL-6 and TNFα cytokines and affects microglial morphology [142]. During chronic inflammation, exosomes modulate microglia towards the M2 phenotype with strong anti-inflammatory and immunoregulatory properties [143].

## 3. EVs in Neurodegenerative Diseases—An Example of ALS

EVs’ important biological roles in intercellular communication in the nervous system make them an interesting entity in neurodegenerative disease progression and pathology.

Amyotrophic lateral sclerosis (ALS) is a heterogeneous, complex disease leading to the collapse of motor functions. Due to its complexity and heterogeneity, it is an overwhelmingly arduous task to develop efficient disease management strategies. TAR DNA-binding protein 43 (TDP-43) was recently identified as the major pathological protein in sporadic ALS and in the most common pathological subtype of FTD, frontotemporal lobar degeneration with ubiquitinated inclusions (FLTD-U). TDP 43 pathology is one of the most striking characters of ALS. Almost 97% of ALS cases (both sporadic and familial) have reported to be positive for TDP-43 pathology [144]. EVs are emerging as a newly discovered mode of intercellular communication in ALS, which provides a deeper insight into ALS propagation mechanisms (Figure 2). Pathological TDP-43 was present in the exosomes isolated from the cerebrospinal fluid (CSF) of ALS patients [145]. Subsequently, the full length and c-terminal fragment of TDP-43 was detected in the EVs from the ALS brain [146]. These EVs had the potential to propagate TDP-43 pathology to healthy cells, thus establishing that the brain may employ EVs to clear toxic TDP-43 [146]. In another interesting study, TDP-43 oligomers were found to be selectively incorporated into EVs that were more potent in propagating TDP-43 pathology than free TDP-43 [147]. Mutated superoxide dismutase 1 gene (SOD1), which is another hallmark molecule of ALS, has also been found in exosomes [148]. Such misfolded SOD1 can be transmitted to other cells via exosomes where it acts as a seed to spread the mutant SOD1 toxicity [149,150]. Thus, the EVs that contained mutant SOD1 were able to transmit toxicity to healthy cells [151]. Neurons are believed to export misfolded SOD1 in exosomes thus getting rid of toxic proteins [152]. Likewise, SOD1 was found to be self-replicating and transferring from one cell to the other [153]. Similar to TDP-43 and mSOD1 propagation, fused in sarcoma (FUS) nuclear DNA/RNA binding protein, and dipeptide repeats of c9orf72 also get transmitted between cells via exosomes [154]. In the presence of c9orf72, exosome secretion is reduced due to decreased interaction between c90rf72 and Rab7L1 [155]. In the serum of ALS patients, mutant SOD1, TDP-43 and FUS were enriched in EV samples [156].

Exosomes have been also implicated to play an important role in the pathobiology of other neurodegenerative diseases as well. Thus, the pathogenic protein Amyloid β peptides, which aggregates in Alzheimer’s disease (AD), was found to be present in exosomes isolated from the plasma and brain of AD patients [157,158]. Another AD pathology marker, protein Tau was found in the exosomes isolated from the CSF of AD patients [159]. Exosomes have been found to propagate the pathological prion protein prpSc that is involved in Creutzfeldt–Jakob disease [160]. Similarly, the α-synuclein protein that is involved in Parkinson’s disease (PD) pathology was detected in exosomes from the CSF of PD patients [161]. Thus, building evidence suggests that EVs play an important role in carrying pathological molecules in neurodegenerative diseases such as ALS, AD, PD and prion diseases.

Apart from proteins, mRNA, miRNA and long non-coding RNA (LncRNA) make for important EV cargo molecules. A different population of miRNA was found in exosomes in ALS that could modulate other mRNA functions [162]. Indeed, miRNA cargo can be used as a diagnostic biomarker and/or disease modifying agent. miRNA secreted into exosomes by astrocytes that harbor c9orf72 mutations disrupts the neuronal network [163]. In the exosomes from CSF of ALS patients, mRNA was found to be differentially expressed suggesting that these biomacromolecules could be used as biomarkers for ALS [164]. Although the presence of LncRNA has not been observed in the exosomes in ALS, these are present in exosomes in other diseases, and it would not be a surprise if it would also be found in the exosomes of ALS.

There is great potential for exosomes to be used as therapeutic agents in ALS. Exosomes have been shown to carry molecules that alter the physiology of recipient cells in ALS. Thus, exosomes derived from adipose stem cells exert neuroprotection to the cells that carry a SOD1 mutation [165]. These exosomes carry a cargo of protein molecules that have antiapoptotic properties by reducing the expression of proapoptotic genes in recipient cells [165]. Similarly, exosomes derived from adipose stem cells improve the coupling efficiency and membrane potential in mitochondria that are diseased due to mutated SOD1 in ALS [166]. Exosomes activate peripheral monocytes to secrete proinflammatory cytokines in ALS [167]. In particular, these exosomes carry TDP-43 that is preferentially taken by peripheral monocytes. Thus, exosomes also positively mediate an important aspect of neuroinflammation in ALS [167].

Although not shown in ALS so far, exosomes can be engineered to deliver therapeutic molecules of choice. Other studies have shown that indeed exosome content can be modified to deliver cargo to recipient cells [168]. This aspect of exosome biology holds a great potential as exosomes can cross the blood–brain barrier [169] and deliver a molecule of choice to the central nervous system. Understanding of exosome neurophysiology is in its infancy also regarding ALS. However, because of their known unique features, exosomes could prove to be a powerful tool to address various aspects of ALS and neurodegenerative diseases in general. Once the contents of exosomes are fully identified, and new techniques of their isolation and purification established, harnessing the power of exosomes is expected to be of tremendous theragnostic value.

## 4. EVs in Axon Regeneration

### 4.1. Peripheral Nerve Injury (PNS) Injury

Trauma or disease that completely severs the axon is a major pathological event both in the peripheral and central nervous system. Distal to the site of interruption, the axon undergoes an active process of degradation, so-called Wallerian degeneration. Restoration of function requires either that the injured axon is able to regrow and achieve functional reinnervation of the target, or that adaptive rearrangements, neuroplasticity (collateral sprouting), occur in appropriate uninjured systems.

Peripheral nerve injury is followed by a complex multicellular and matrix response [170]. The combined actions of immune cells, fibroblasts, vascular cells and Schwann cells form an environment at the injury site and along the distal stump that promotes axonal growth. Schwann cells and associated axons maintain a close and coordinated interaction from early development and throughout life. One important aspect of this interaction is the continuous exosomal transfer from Schwann cells to the axon of molecules (RNAs, lipids and proteins/peptides), which are vital for maintaining functional properties of the axon, and for allowing its remodeling in response to altered circumstances [171,172,173]. Through this subcellular transfer system, the axon obtains some degree of independence from the nerve cell body in a short-term perspective. Such independence is presumably of particular relevance for long axons where molecular cargos originating from the cell body would require days to reach the distal parts of the axon (sensory axons and motor neuron axons can be up to 1 m long).

Not surprisingly, Schwann cell-derived EVs play an important role in the elongation of injured peripheral nerve axons by contributing to rebuilding the axon with structural and regulatory molecules [174,175]. Recent findings indicate that microRNAs (miRNAs) carried in exosomes from Schwann cells, macrophages and neurons have a significant impact in the processes that underlie peripheral nerve regeneration [176]. Proteomic analysis of Schwann cell-derived exosomes revealed the presence of several proteins and signaling pathways that are involved in axon regeneration [177]. Inflammatory cells appear also to promote the emergence of a growth promoting state in the injured neuron through the release of exosomal NADPH oxidase 2 (NOX2) complexes into the injured axons. These are converted to active NOX2, which is retrogradely transported to the cell body, where it oxidizes and inactivates the axon growth inhibitor phosphatase and tensin homolog (PTEN), thereby promoting axonal outgrowth [178].

The growth promoting effect from mesenchymal stem cell (MSC) transplants on peripheral nerve regeneration appears also to depend on the exosomal transfer of growth promoting molecules [179,180,181,182,183,184,185]. Recent findings also indicate that axonal pathfinding after an injury can be influenced by exosomes from denervated (but not naïve) muscle [186]. Thus, the exploitation of exosomes with optimized molecular content may help to overcome the major challenges in clinical peripheral nerve repair: (i) bridging long gaps at the injury site; (ii) achieving functional reinnervation of targets distant from the injury and (iii) directing regenerating axons to their appropriate target [187]. Moreover, recent findings show that intrathecal administration of MSC-derived exosomes attenuate symptoms in the nerve injury-induced animal model of neuropathic pain [188].

### 4.2. CNS Injury

In contrast to the potential for successful regeneration and target reinnervation in the peripheral nervous system, axon injury in the central nervous system faces the emergence of a growth inhibitory environment in combination with a decline in the regenerative capacity of injured neurons [189,190]. However, the regeneration on the tissue level in the CNS is well documented [191]. Therefore, multiple and diverse targets have to be influenced in order to overcome regeneration failure and to induce functional regeneration after CNS injury.

Retinoic acid (RA) signaling has been shown to mediate intrinsic beneficial effects on spinal cord repair. Thus, the activation of neuronal RA receptor β (RARβ) inactivates the growth inhibiting effect of PTEN, and induces the release of exosomes. These are subsequently taken up by astrocytes and counteract their glial scar formation [192] allowing isolation of damaged parts of the CNS from the healthy uninjured areas. In addition, RARβ activation leads to the synthesis of RA in NG2-glia and subsequent exosome-associated further RA release [193,194]. At the same time myelin-associated growth inhibitors are released through exosomes from oligodendrocytes [195]. Therefore, active production of exosomes in the injuries areas of CNS may have a strong effect on CNS neuroplasticity, including collateral sprouting and unmasking of latent synapses [196,197].

MSC-derived EVs have been found to promote spinal cord injury repair in animal models through different processes. Exosomal delivery to the spinal cord, locally or systemically, induces axonal growth [198,199], counteracts neuroinflammation [200,201,202], reduces glial scar formation [203], downregulates harmful astroglial properties [204,205], attenuates neuronal apoptosis [206], promotes angiogenesis [200,207] and protects the blood–spinal cord barrier [208]. Intranasal administration of MSC-derived exosomes containing small interfering RNA (siRNA) for PTEN have been successfully used to stimulate spinal cord injury repair by downregulating PTEN [209]. Beneficial effects after spinal cord damage have also been demonstrated after genetic [206,210,211,212] or extracellular [210] modification of MSC properties prior to obtaining their exosomes.

These findings indicate that exosomes have the potential to contribute to CNS injury repair by modifying the hostile environment and promote neuronal growth capacity.

## 5. Neuroprotective Properties of EVs from the Dental Pulp Stem Cells

The potential use of EVs for therapeutic purposes attracted a great deal of interest in recent years. This approach has several advantages. First, it does not require the direct use of cells avoiding and reducing their limitations and risks associated with cell transplantation. Second, EVs are relatively simple systems that are more suitable for the large-scale clinical manufacturing. Third, in contrast to most pharmaceutical products EVs can easily cross the blood–brain barrier (BBB) and enter the brain, thus eliminating the need of neurosurgical interventions [211,212]. Moreover, several reports demonstrated therapeutic efficacy of minimally invasive intranasal administration of EVs in different animal models of neurological disorders [213,214]. On the other hand, this therapeutic approach may be challenging, because the molecular composition of EVs is complex and it is very difficult to precisely identify molecules responsible for the therapeutic effect. We therefore suggest that in most cases therapeutic properties of EVs depend on the net effect of different cargo molecules, rather than one “magic bullet”. From the therapeutic perspective, mesenchymal stem cells (MSCs) represent one of the most promising sources of EVs. Here we will briefly summarize the current state of knowledge about neuroprotective properties of the dental pulp-derived MSCs. Dental pulp stem cells derived from permanent teeth (DPSCs), or stem cells derived from the dental pulp of human exfoliated deciduous teeth (SHEDs) originate from the cranial neural crest cells, which are precursors of both neural and skeletal tissues [215]. Genetic lineage tracing studies revealed that DPSCs originate from peripheral nerve-associated glia [209] and from periarterial cells in the neurovascular bundle of dental pulp [216]. Thus, in contrast to MSCs from other mesodermal tissues such as bone marrow or adipose tissue, DPSCs could be particularly suitable for the induction of neural differentiation and also for the neurorestorative therapies. After transplantation these cells can integrate into the CNS, but their survival rates are very low. Nevertheless, most studies reported considerable therapeutic improvements. This apparent paradox could be explained by a paracrine action of DPSCs and SHEDs. Indeed, several studies used a conditioned medium from DPSCs or SHEDs to demonstrate neuroprotective action in vitro [217] and in vivo [218]. Apart from soluble molecules the secretome of cultured cells contains different types of membrane-enclosed EVs. We investigated whether EVs derived from SHEDs display neuroprotective properties during 6-OHDA-induced oxidative stress in human dopaminergic neurons [219]. We found that exosomes, but not microvesicles derived from the microcarrier cultures of SHEDs suppressed 6-hydroxydopamine (6-OHDA)-induced apoptosis in dopaminergic neurons. Interestingly neuroprotective (antiapoptotic) effects have only been observed when exosomes isolated from SHEDs cultivated on microcarriers in the bioreactor were used, but not in the case of exosomes from standard SHED cultures indicating that culture conditions may dramatically affect molecular composition and therapeutic properties of EVs [220]. Exosomes secreted by SHEDs also reduced the neuroinflammatory response in BV2 microglial cells [221]. We have recently demonstrated that EVs derived from SHEDs suppressed LPS-induced activation of the NFκB signaling pathway and upregulated phagocytic activity of human microglial cells [222]. As mentioned, EVs can enter the CNS after intravenous administration, therefore neurosurgical procedures become unnecessary [223]. On the other hand, pharmacokinetic studies in vivo have shown that EVs are very quickly removed from the bloodstream and most of them are entrapped in the liver and lungs [224]. Accordingly, several groups investigated alternative delivery via the minimally invasive intranasal route [225]. Importantly, after intranasal application exosomes selectively accumulated in the affected areas of the brain [226]. For example, after intranasal application to the 6-OHDA-treated mice (Parkinson‘s disease (PD) model), exosomes selectively accumulated in the damaged striatum areas even up to 96 h [226]. We have investigated the therapeutic effects of intranasal administration of EVs derived from human SHEDs on the unilateral 6-OHDA medial forebrain bundle (MFB) rat model of PD [227]. We found that intranasal therapy with EVs significantly improved the motor function of PD affected rats, and also normalized the expression of tyrosine hydroxylase in the SN and striatum areas [228]. Our studies for the first time demonstrated therapeutic efficacy of EVs in the 6-OHDA rat PD model. More recently we investigated the effects of intranasally administered EVs on the cognitive dysfunctions that occur after unilateral injection of 6-OHDA into the MFB of experimental rats [229]. We also compared short- and long-term therapeutic effects of EVs on 6-OHDA-induced gait and cognitive disturbances. Our data indicate that EVs could effectively reverse spatial learning/memory impairments caused by intra-MFB injection of 6-OHDA. These effects occurred shortly (up to 5 days) after treatment discontinuation. The reversal of 6-OHDA-induced gait disturbances by EVs was even longer—for up to 10 days after the end of their administration [230]. Thus, our data demonstrate the temporary nature of the therapeutic action of EVs during 6-OHDA-induced neurodegeneration and suggest that long-term treatment could be necessary to maintain gait and cognitive improvements. We expect that these findings could be potentially exploited for the development of new treatment strategies against PD.

In conclusion DPSCs and SHEDs represent a very promising source of EVs that can be potentially used for the treatment of different neurological disorders. However, for therapeutic applications large amounts of EVs will be needed. These amounts can only be obtained by using new technologies allowing scaling up production and isolation of EVs [29,231].

## 6. Discussion

Stem cells, already for at least two decades, have become one of the most attractive tools to treat the nervous tissue affected by ischemia, degeneration or trauma. This type of therapy holds great promise and the progress in this field is substantial. On the other hand, our knowledge of mechanisms, which are present in the background of observed benefits, is still scarce. It became obvious that an insufficient level of understanding exists of the mechanisms and methods of tissue repair related to multifactorial elements present in this complex process. Observed repair and neuroprotection effects described in numerous independent studies suggest that in both preclinical and clinical trials positive effects are established with secreted growth factors, and EVs.

In the present work a detailed overview of the role of EVs in neural development and regeneration is presented and resumed on Table 1.

The origin of the already well-proven effects of a specific type of EVs, i.e., exosomes, can be found already during the embryonic development. Indeed, the example of EVs represents a well-known link between tissue regeneration after injury/degeneration and genetic pathways present during embryo development. The missing link between reactivation of proregenerative signals and recovery, for instance, after a brain injury can be investigated through the secretome of the cells during those two processes. This has been elegantly shown in both examples of recovery after a brain injury or brain ischemia. Likewise, a combination of molecules that regulate the outgrowth of axons are thus again present in the secretome after stem cell reactivation. It is important to add that EVs are secreted by all cells of the nervous system and that their roles are different. While microglial EVs can possess both positive and negative effects on regeneration, astrocytic exosomes, which there is still not enough known about them, possess a strong regulatory action. Those secreted by neuronal precursors are shown to have the strongest regenerative potential.

Here the latest advancements in the discovery of the enrichment of various RNAs derived from systemic injection of exosomes were also discussed in detail. They induce neurological recovery and promote neuroregeneration by achieving a critical mass of cargo proteins and non-coding RNAs. It is important to notice that those secretomes are heterogeneous and still a lot of work is needed to understand if those differences are linked to innate heterogeneity of a cellular response or they are fine-tuned in the dependence of different pathological processes.

Although extensive evidence from preclinical studies shows that secretome and vesicles derived from stem cells represent valuable therapeutic options for the treatment of neuronal pathology, it is needed to recognize challenges and address them properly with the uppermost and detailed care prior to their clinical application. This is especially needed from a regulatory point of view. Even though experiments in the laboratory setup yielded numerous conclusions that EVs bring benefits, production of clinically graded products ask for stringent and precise standardization. Based on the present detailed insight into the current state of the art, the next important steps toward the clinical application of EVs might include the determination of EV features in terms of size and properties, alongside with the identification of concentration ranges for theragnostic purposes.

## Figures and Tables

**Figure 1 ijms-21-06859-f001:**
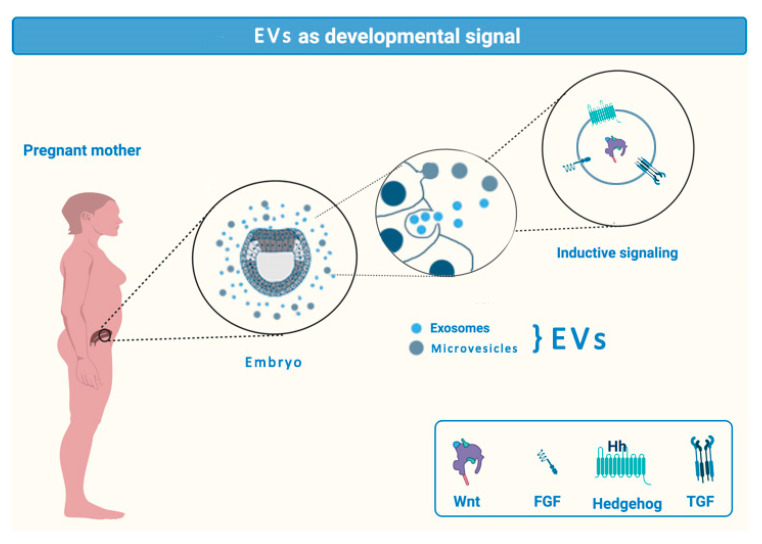
Exosomes as developmental signals.

**Figure 2 ijms-21-06859-f002:**
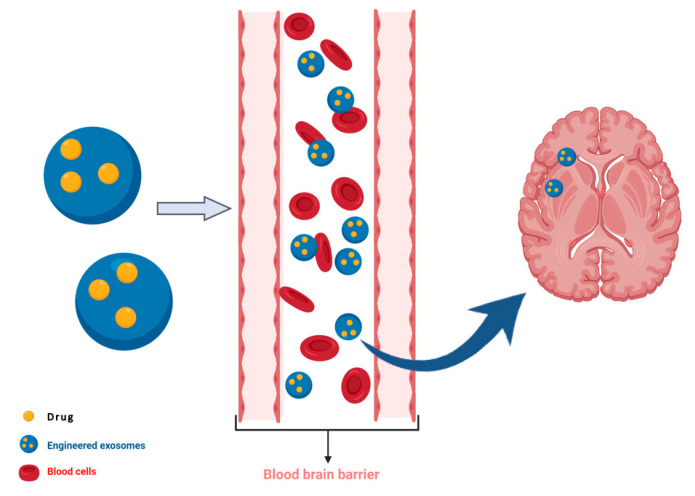
Extracellular vesicles (EVs) could be engineered in order to pass the blood–brain barrier and treat neurodegenerative diseases.

**Table 1 ijms-21-06859-t001:** Functions of EVs in neuroinflammation/neurodegeneration.

Role of EV	Activity	Ref
Central Nervous System	Crucial role in neurorestoration after ischemic injury	[139,140]
	Induced exosomes (iEXOs) secreted from induced neural progenitor cells (iNPCs) stimulated proliferation of neural progenitor cells (NPCs)	[141]
	During chronic inflammation, exosomes modulate microglia towards the M2 phenotype with strong anti-inflammatory and immunoregulatory properties	[140,141,142,143]
	Myelin-associated growth inhibitors are released through exosomes from oligodendrocytes	[195]
	Active production of exosomes in injured areas of CNS may have a strong effect on neuroplasticity, including collateral sprouting and unmasking of latent synapses	[196,197]
	Exosomes secreted by SHEDs reduced neuroinflammatory response in BV2 microglial cells	[221]
	EVs derived from SHEDs suppressed LPS-induced activation of the NFκB signaling pathway and upregulated phagocytic activity of human microglial cells	[222]
Amyotrophic lateral sclerosis (ALS)	Pathological TDP-43 was present in the exosomes isolated from the CSF of ALS patients. These EVs had the potential to propagate TDP-43 pathology to healthy cells	[145,146,147]
	Mutated SOD1 gene, a hallmark molecule of ALS, is present in exosomes and is thus transmitted to other cells	[148,149,150,151,152,153,154,155,156]
	EVs that contained mutant SOD1 were able to transmit toxicity to healthy cells	[151]
	A population of miRNA was found in exosomes that could modulate mRNA function.	[162,163,164]
	Exosomes derived from adipose stem cells improve the coupling efficiency and membrane potential in mitochondria that are diseased due to mutated SOD1	[166]
	Exosomes activate peripheral monocytes to secrete proinflammatory cytokines in ALS	[167]
Alzheimer’s disease (AD)	The pathogenic Amyloid β peptides are present in exosomes isolated from plasma and brain of AD patients	[157,158]
	Protein Tau, an AD pathology marker, was found inexosomes isolated from patients’ CSF	[159]
Creutzfeldt–Jakob disease	Exosomes can propagate the pathological prion protein prpSc	[160]
Parkinson’s disease (PD)	Proteins related to this pathology were detected in exosomes from the CSF	[161]
Peripheral Nerve Regeneration	Schwann cells transfer exosomes to the axon in order to transfer vital molecules for structural plasticity (RNAs, lipids and proteins/peptides)	[171,172,173,176,177]
	Schwann cell-derived EVs help in elongation of injured peripheral nerve axons by contributing to rebuilding the axon with structural and regulatory molecules	[174,175]
	The growth promoting effect from MSC transplants on peripheral nerve regeneration depends on the exosomal transfer of growth promoting molecules	[179,180,181,182,183,184,185]
Effects of exosomal delivery to the spinal cord, locally or systemically:	Induces axonal growth	[198,199]
	Counteracts neuroinflammation	[200,201,202]
	Reduces glial scar formation	[203]
	Downregulates harmful astroglial properties	[204,205]
	Attenuates neuronal apoptosis	[206]
	Promotes angiogenesis	[200,207]
	Protects the blood–spinal cord barrier	[208]
	Stimulates spinal cord injury repair by downregulating PTEN	[209]
	Exosomes, derived from the microcarrier cultures of SHEDs suppressed 6-OHDA-induced apoptosis in dopaminergic neurons.	[220]

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
