# Peer review of "Extracellular Vesicles as Innovative Tool for Diagnosis, Regeneration and Protection against Neurological Damage"

_ijms, 2020, doi:10.3390/ijms21186859_

Round 1

Reviewer 1 Report

The authors review the role of extracellular vesicles (EVs) as innovative tool for diagnosis, regeneration and protection against neurological damage. The manuscript is well organized and the topic is of great interest for the scientific community since extracellular vescicles represent at the moment one of the more interesting tool for new treatments in neurodegeneration. 

Minor revision:

1) For the benefit of the clarity and to make the informations reported more easy for the readers,  the authors should summarize the functions of EVs in neuronflammation/neurodegeneration in a Table. 

2) Some references are missing. Campanella C et al., and Caruso Bavisotto et al, have discussed the choise  of EVS for therapeutic purposes and the involvement of EVs in the cell-cell communication in the nerrvous system (2019). 

Author Response

Dear Reviewer

thanks for your work that strongly improve the review. Following your suggestion we added novel references and a table that summarize the functions of EVs in neuronflammation/neurodegeneration . 

Reviewer 2 Report

Andjus et al. present a nice review manuscript about microvesicles. However, there are few concerns:

  1. This kind of review papers have are massively used as educational material and therefore should be structured to deliver new knowledge. In line 87 'classical' exosomes terminology is used without any prelude. Please include a sentence or two describing 'classical' exosomes. 
  2. Citation (28) for the sentence starting in line 121 is not enough. First of all, that citation does not cover all 3 (paracrine, endocrine, and autocrine) modes of exerting their functionality. And also, having only 1 citation for such a broad statement might force the readers to have an impression that only cancer cells can communicate in such way. 
  3. Please revise all of your citations. 
  4. A good portion of the manuscript discusses the role of microvesicles in embryology. Perhaps, authors need to revise the title of the manuscript or revise the manuscript itself to drag the focus to neurology by moving embryology section to introduction.

Author Response

Dear Reviewer

thanks for your work that strongly improve the review. As request we:

  1.  In line 87 'we included a sentence or two describing 'classical' exosomes. 
  2. We increased the citation  covering modes of exerting their functionality. 
  3. We revised all of our citations. 
  4. We  moved embryology section to introduction.